

# Non-destructive prediction of anthocyanin concentration in whole eggplant peel using hyperspectral imaging

Zhiling Ma[1,*], Changbin Wei[1,*], Wenhui Wang[1], Wenqiu Lin[1], Heng Nie[1], Zhe Duan[1,2], Ke Liu[1,3] and Xi Ou Xiao[1]

[1] South Subtropical Crop Research Institution, Chinese Academy of Tropical Agricultural Sciences, Key Laboratory of Tropical Fruit Biology, Ministry of Agriculture & Rural Affairs, Key Laboratory of Hainan Province for Postharvest Physiology and Technology of Tropical Horticultural Products, Academy of Tropical Agricultural Sciences, Zhanjiang Key Laboratory of Tropical Crop Genetic Improvement, Zhanjiang, Guangdong, China
[2] Yunnan Agricultural University, Puer, Yunnan, China
[3] South China Agricultural University, Guangzhou, Guangdong, China
[*] These authors contributed equally to this work.

Corresponding author
Xi Ou Xiao,
xiao-forlearning@163.com

## ABSTRACT

Accurately detecting the anthocyanin content in eggplant peel is essential for effective eggplant breeding. The present study aims to present a method that combines hyperspectral imaging with advanced computational analysis to rapidly, non-destructively, and precisely measure anthocyanin content in eggplant fruit. For this purpose, hyperspectral images of the fruits of 20 varieties with diverse colors were collected, and the content of the anthocyanin were detected using high performance liquid chromatography (HPLC) methods. In order to minimize background noise in the hyperspectral images, five preprocessing algorithms were utilized on average reflectance spectra: standard normalized variate (SNV), autoscales (AUT), normalization (NOR), Savitzky–Golay convolutional smoothing (SG), and mean centering (MC). Additionally, the competitive adaptive reweighted sampling (CARS) method was employed to reduce the dimensionality of the high-dimensional hyperspectral data. In order to predict the cyanidin, petunidin, delphinidin, and total anthocyanin content of eggplant fruit, two models were constructed: partial least squares regression (PLSR) and least squares support vector machine (LS-SVM). The HPLC results showed that eggplant peel primarily contains three types of anthocyanins. Furthermore, there were significant differences in the average reflectance rates between 400–750 nm wavelength ranges for different colors of eggplant peel. The prediction model results indicated that the model based on NOR CARS LS-SVM achieved the best performance, with a squared coefficient of determination ($R^2$) greater than 0.98, RMSEP and RMSEC less than 0.03 for cyanidin, petunidin, delphinidin, and total anthocyanin predication. These results suggest that hyperspectral imaging is a rapid and non-destructive technique for assessing the anthocyanin content of eggplant peel. This approach holds promise for facilitating the more effective eggplant breeding.

## INTRODUCTION

Anthocyanins have a C6-C3-C6 skeleton as their structural basis and are often glycosylated (*Castañeda Ovando et al., 2009*). The characteristics of hydroxylated groups, the types and the number of bonded sugars to their structure; the aliphatic or aromatic carboxylates linked to the sugar in the molecule, and their positions have collectively led to the identification of approximately 23 anthocyanidins (*Castañeda Ovando et al., 2009*). In plants, only pelargonidin, cyanidin, peonidin, malvidin, petunidin, and delphinidin have been detected (*de Pascual-Teresa & Sanchez-Ballesta, 2007*). Apart from showcasing a range of vibrant colors, the anthocyanins also play a crucial role in the plant's stress response (*Kaur et al., 2023*; *Naing & Kim, 2021*; *Yan et al., 2022*). The augmentation of anthocyanin content in plants has garnered increasing attention from researchers as a means to enhance plant quality and resilience against biotic and abiotic stress (*Kaur et al., 2023*; *Li & Ahammed, 2023*). More importantly, the anthocyanins have important implications in the field for improving human health (*Speer et al., 2020*; *Tsuda, 2012*).

Eggplant (*Solanum melongena* L) is an economically important vegetable that is widely grown worldwide. The fruit of eggplant is rich in phenolic compounds such as anthocyanin, chlorogenic acid and vitamin P, all of which are beneficial for human health (*Basuny, Arafat & El-Marzooq, 2012*; *Dong et al., 2020*; *Plazas et al., 2013*; *Todaro et al., 2009*). Because of its high content of phenolics, eggplant has been classified among the top ten vegetables with antioxidant capacity (*Niño Medina et al., 2017*). The color of the eggplant fruit varies from white, green, purple, and dark-purple, depending on the type and content of anthocyanin (*Niño Medina et al., 2017*). Several results indicate that the eggplant peel anthocyanins contain delphinidin, petunidin, malvidin, and cyanidin depending on the eggplant variety and anthocyanins extraction methods (*Basuny, Arafat & El-Marzooq, 2012*; *Ferarsa et al., 2018*; *Niño Medina et al., 2017*; *Nothmann, Rylski & Spigelman, 1976*). Currently, there were two primary methods for measuring the anthocyanin content of the eggplant peel: HPLC for individual anthocyanin analysis and the pH differential method for total monomeric anthocyanin quantification (*Ferarsa et al., 2018*; *Zhang et al., 2014*). Both methods require grinding the sample and extracting anthocyanins, a process that consumes several hours and is time-consuming and labor-intensive. Therefore, it is difficult to measure anthocyanin content on a large scale. For example, to map the QTLs that regulate the anthocyanin biosynthesis, the anthocyanin content was only detected by visual discrimination based on color as the samples are often several hundred or even thousands (*Guan et al., 2022*; *Toppino et al., 2020*). Furthermore, these methods are destructive and result in the production of chemical residues. Thus, it is crucial to establish an efficient method for measuring anthocyanins content and type.

Hyperspectral imaging is a high-throughput method used for analyzing plant phenotypes, including abiotic, biotic, and chemical properties testing (*Sarić et al., 2022*). Due to its advantages in high-throughput and non-destructive detection, it excels in chemical property testing, particularly in the analysis of anthocyanins, and has been widely utilized (*Caporaso et al., 2018*; *Chen et al., 2015*; *Dai et al., 2023*; *Fernandes et al., 2011*; *Hernández-Hierro et al., 2013*; *Li et al., 2023*; *Pandey et al., 2017*; *Qin & Lu, 2008*;

*Tian et al., 2020*; *Yang et al., 2015*; *Zhang et al., 2017*). For example, *Zhang et al. (2017)* found that the squared correlation coefficient ($R^2$) and root mean square error (RMSE) for anthocyanins in wine grape skins reached 0.87 and 0.1442 (g/L M3G), respectively. *Yang et al. (2015)* demonstrated that the optimal predictive model for quantifying anthocyanins in lychee pericarp during storage achieved an $R^2$ value of 0.896 and RMSE of 0.567%. The SAE-GA-ELM-based model used to predict anthocyanin content in mulberry fruits achieved the best performance, with an $R^2$ of 0.97 in the training dataset and an RMSE of 0.22 mg/g in both the training and testing datasets (*Li et al., 2023*). These findings indicate that hyperspectral imaging technology can be used for non-destructive detection of plant anthocyanins in fruit peels.

In the present study, we aim to develop a prediction model that connects hyperspectral imaging with anthocyanin content for non-destructive detection in eggplant peels. This research provides a foundation for eggplant breeding and may have significant implications for future studies.

## MATERIALS AND METHODS

### Plant material

A total of 20 eggplant varieties bred by the Chinese Academy of Tropical Agricultural Sciences, South Subtropical Crop Research Institute were selected. These 20 eggplant varieties encompass a range of colors including white, green, light-purple, green-purple, and dark-purple. In total, 277 eggplant fruits were collected, consisting of 10 samples of white color, 28 samples of green color, 111 samples of light-purple color, 17 samples of green-purple color, and 111 samples of dark-purple color (Table S1).

### Hyperspectral image acquisition

The imaging system comprised of a SOC70VP hyperspectral camera and a lamp holder. The hyperspectral camera covered wavelengths ranging from 400 to 1,000 nm with an approximate resolution of 0.6 nm with 128 pixels (channels) in the wavelength dimension. The lamp holder accommodated two Philips halogen lamps with a power rating of 500 Watts at 220 volts. The eggplant and spectralon were positioned beneath the hyperspectral camera to allow reflection of the light emitted by the halogen lamps.

Following hyperspectral imaging, one mm uniformly thick peel was collected using a sharp knife immediately, frozen using liquid nitrogen, and stored at −80 °C for subsequent anthocyanin content analysis. Reflectance values were calculated using the SRAnal 710 software. ROI extraction and the calculation of the average reflectance were performed using ENVI 5.3.

### Anthocyanin extraction

The identification and quantification of anthocyanins in eggplant peel were conducted using High Performance Liquid Chromatography (HPLC) methods. Six types of anthocyanins were detected, namely delphinidin, cyanidin, petunidin, pelargonidin, paeonidin, and malvidin. The extraction and hydrolysis methods for anthocyanins were adjusted based on the procedure outlined in HPLC(2014). The extraction process involved a solution

consisting of anhydrous ethanol, water, and hydrochloric acid in a ratio of 2:1:1. To initiate the extraction, 1.0 g of powder was accurately weighed and transferred into a 10 mL volumetric flask with a stopper. The extractant was then added to the mark, and the mixture was vigorously shaken for 1 min. Ultrasonic extraction was subsequently performed under light-protected conditions for 30 min. For the hydrolysis of anthocyanins to anthocyanidins, the extract obtained from ultrasonic extraction underwent a boiling water bath for 1 h. After cooling, additional extractant was added to bring the total volume to 10 mL. The mixture was thoroughly shaken and allowed to settle. The supernatant was collected and filtered through a 0.22 μm organic membrane.

## HPLC analysis of anthocyanins

The HPLC analysis followed the reference method for determining anthocyanidins in plant origin products-High performance liquid chromatography in *Deineka & Grigor'ev (2004)*. The HPLC system utilized for the analysis was the LC-20A equipped with a UV detector. A mobile phase A consisting of formic acid and water in a ratio of 1:9 was used for the HPLC analysis. Meanwhile, mobile phase B consisted of methanol, acetonitrile, water, and formic acid in a ratio of 22.5:22.5:40:10. The analysis employed a gradient elution method with specific time intervals and percentages of mobile phase B as follows: 0–2 min: 7–40% B; 2–11 min: 40–67% B; 11–12 min: 67–100% B; 12–14 min: 100% B; 14–15 min: 100–7% B; 15–20 min: 7% B. Each sample was injected with a volume of 10 μl, and three replicates were performed for each sample.

The standards for delphinidin, cyanidin, petunidin, pelargonidin, peonidin, and malvidin were acquired from Sigma. The total content of anthocyanin was determined by summing the quantities of these six types of anthocyanins.

## Hyperspectral data preprocessing

To reduce the influence of factors such as background noise on the average reflectance, pre-processing was performed on the average reflectance data, as shown in Table 1 using Matlab 2020a (The MathWorks Inc., Natick, MA, USA). Following pre-processing, the data were divided into a training dataset and a test dataset at a ratio of 7:3 using the randperm function.

## Feature variables extraction

To reduce the number of input variables and improve the model efficiency, the spectral data features were extracted by CARS as described by *Li et al. (2009)* with the default parameters.

## Modeling algorithms

The LS-SVM were implemented using Matlab 2020a with LS-SVMlabv1_8. PLSR were completed by using Matlab 2020a. The flowchart of PLSR analysis is shown in Fig. 1. The following statistical parameters were calculated:

**Table 1  The pre-processing of average reflectance.**

| | Preprocessing method |
|---|---|
| 1 | Non-Preprocessing |
| 2 | SNV (Standard Normalized Variate) |
| 3 | AUT (Autoscales) |
| 4 | NOR (Normalize) |
| 5 | SG (Savitzky-Golay Convolutional) |
| 6 | MC (Mean Centering) |

root mean square error of prediction (RMSEP).
root mean square error of calibration (RMSEC).

$$RMSE = \sqrt{\frac{1}{n}\sum_{i=1}^{n}(Measurement_i\text{-}Predicated_i)^2}$$

$R^2_P$ (coefficient of determination of Prediction).
$R^2_C$ (coefficient of determination of Calibration).

$$R^2 = 1 - \frac{\sum_{i=1}^{n}(Meansurement_i - Predicated_i)^2}{\sum_{i=1}^{n}(Measurement_i - Mean(Meantsurement))^2}$$

RPDp (Ratio of standard deviation of the validation set to standard error of prediction of Prediction).

RPDc (Ratio of standard deviation of the validation set to standard error of prediction of Calibration).

$$RPD = \frac{SD}{RMSEP}.$$

# RESULTS

## Various eggplant varieties have differing levels of anthocyanin content and demonstrate distinct average reflectance spectra

The results indicate that eggplants of different colors contain varying types of anthocyanins (Figs. 2A and 2B, Table S1). The peels of 138 eggplants contained cyanidin and delphinidin, while the peels of 67 eggplants contained petunidin and delphinidin. Additionally, 21 eggplants contained cyanidin, delphinidin, and petunidin. Notably, none of the 277 eggplants analyzed contained pelargonidin, peonidin, or malvidin. Except for two white eggplants that contained 0.9834 μg/g and 0.6368 μg/g of delphinidin, the remaining white and green eggplants had undetectable anthocyanin content. Cyanidin content ranged from 0 to 9.7430 μg/g, delphinidin content ranged from 0 to 660.177 μg/g and petunidin content ranged from 0 to 17.3905 μg/g. The total anthocyanins content ranged from 0 to 668.049 μg/g.

The average reflectance varied among eggplants of different colors. Both the green and green-purple fruits exhibited an average reflectance model characteristic of green plants, with a peak at 550 nm. However, the green-purple eggplant fruit had lower reflectance

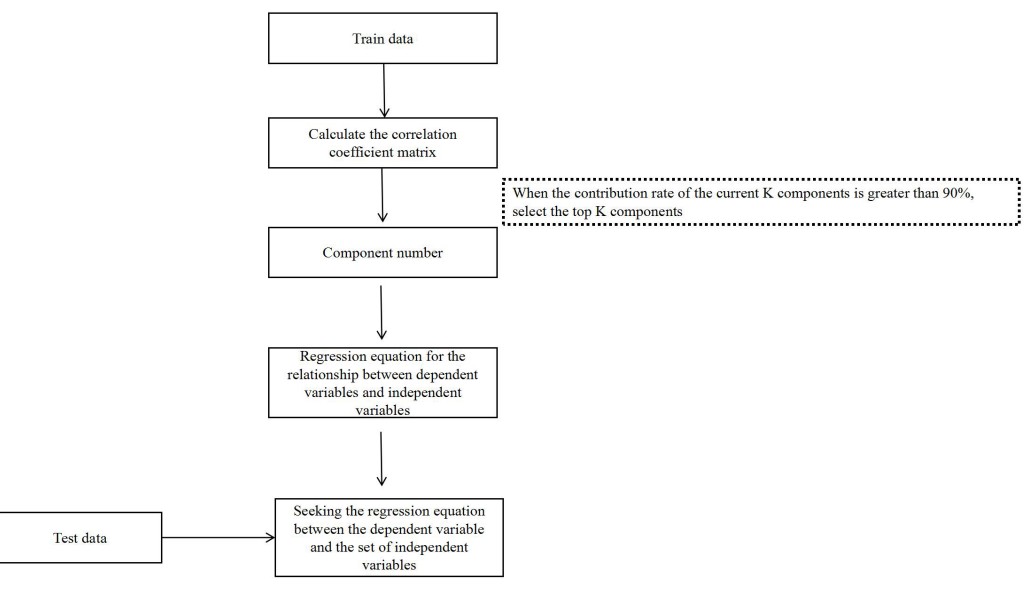

**Figure 1** **The flowchart of PLSR analysis.**

compared to the green eggplant fruit due to the presence of anthocyanins in the green-purple variety. White eggplants demonstrated the highest reflectance between 400–700 nm, while dark-purple eggplants had the lowest reflectance. Light-purple and dark-purple eggplants displayed a minimum reflectance around 500 nm (Fig. 2C).

## Hyperspectral data preprocessing analysis

In order to establish reliable prediction models, we applied five pretreatment methods to the spectral data and extracted the feature variables using CARS. The results indicated that the average reflectance became more concentrated after pretreatment, compared to the non-preprocessed reflectance (Fig. 3).

To enhance the accuracy and robustness of the diagnostic models, the CARS were utilized to extract feature variables from the pool of 128 variables. In the case of petunidin, only five feature variables were present without any preprocessing. However, after pretreatment, the number of feature variables increased to 128 for SNV, six for AUT, five for NOR, 12 for SG, and four for MC pretreatment, as shown in Table S2 and Fig S1. Similarly, in the case of cyanidin, there were initially only five feature variables without any preprocessing. However, after pretreatment, the number of feature variables increased to 128 for SNV, six for AUT, five for NOR, 12 for SG, and four for MC pretreatment, as indicated in Table S2 and Fig S2. Likewise, for delphinidin, the number of feature variables was initially 5 without any preprocessing. However, following pretreatment, the number increased to 128 for SNV, six for AUT, 24 for NOR, 20 for SG, and five for MC pretreatment, as provided in Table S2 and Fig S3. Finally, in the case of total anthocyanins, the initial count of feature variables without any preprocessing was five. However, after pretreatment, the number increased to 128 for SNV, 25 for AUT, nine for NOR, 32 for SG, and 15 for MC pretreatment, as described in Table S2 and Fig S4.

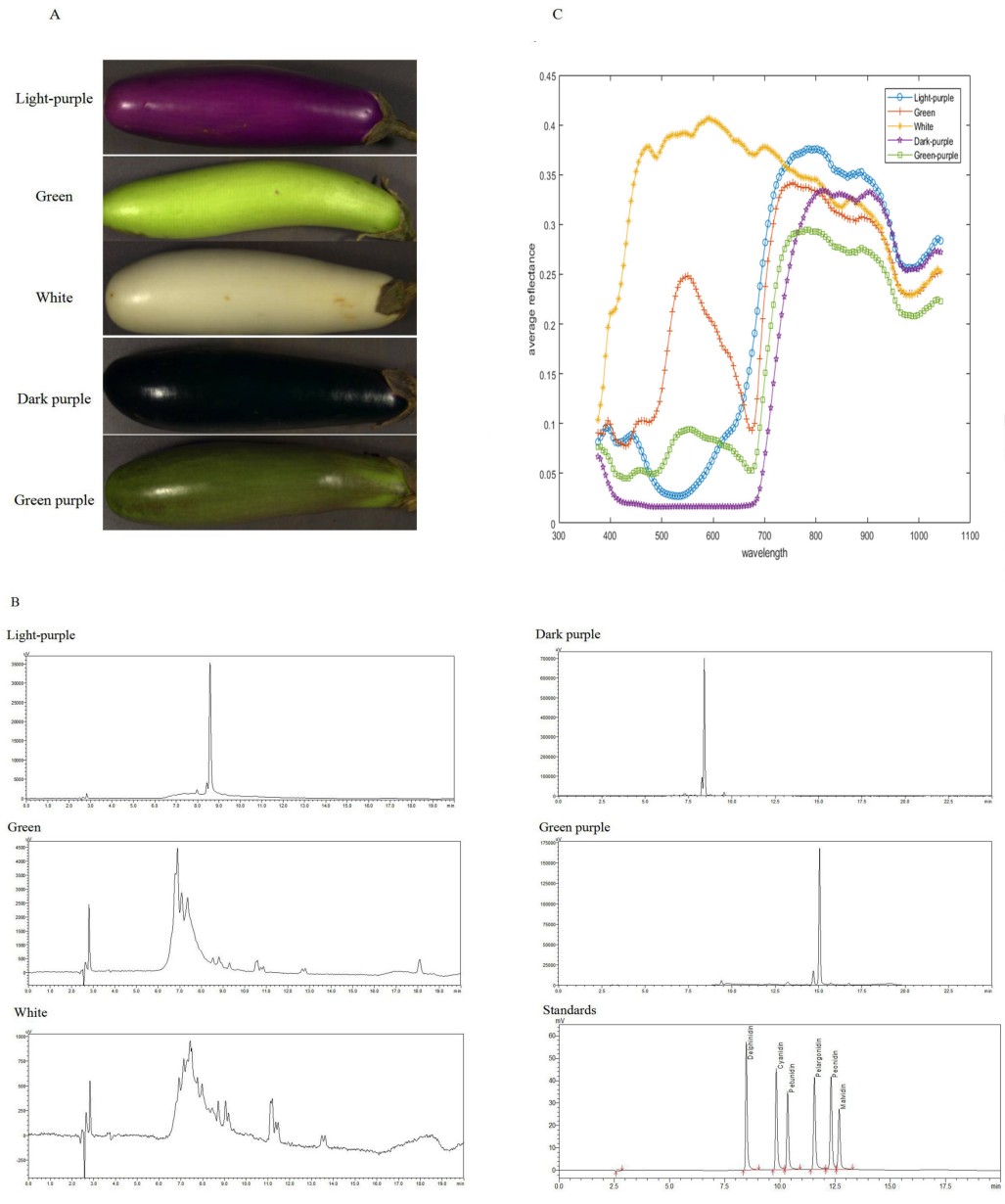

**Figure 2** **The HPLC chromatogram and spectral reflectance of eggplant peel with different colors.** (A) The eggplant peel with different colors. (B) The HPLC chromatogram of eggplant peel with different colors. (C) The spectral reflectance of eggplant peel with different colors.

## Modeling and validation of the regression models

In this study, the PLSR and LS-SVM models were utilized to develop estimation models for cyanidin, delphinidin, petunidin, and total anthocyanins. The performance of the PLS regression model on the cyanidin was evaluated. Despite the SNV-PLSR model having the highest $R_c^2$ (0.3604) and $R^2_P$ (0.3825), the respective RMSEC and RMSEP values were 3.3993 and 3.6949 (Fig. 4). This result indicates a significant disparity between the measured

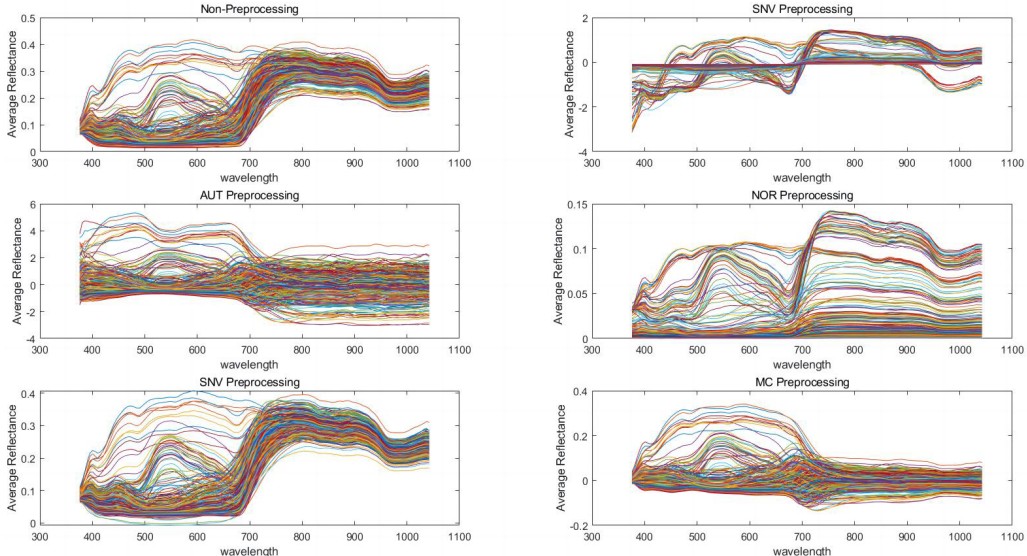

**Figure 3 Preprocessing of spectral data with different algorithms.** Standard normalized variate (SNV), autoscales (AUT), normalization (NOR), Savitzky-Golay convolutional smoothing (SG), and mean centering (MC).

and predicted cyanidin content. Interestingly, the LS-SVM model demonstrated superior predictive performance compared to the PLSR model. However, the perfect predictive outcome of the SNV LS-SVM model may be unreliable due to warnings from Matlab software suggesting potential issues with singularity or improper scaling, thereby resulting in inaccurate predictions. The NOR preprocessing LS-SVM model achieved the best results with with an $R_c^2$ value of 9955, and $R_p^2$ value of 0.9918. Additionally, the RMSEC and RMSP values were 0.0183 and 0.0275, respectively, The RPDc and and RPDp values were 14.5478 and 10.91075, respectively (Fig. 5).

To further enhance prediction accuracy, feature variables extracted by CARS were used to build the LS-SVM model. Similar caution should be exercised regarding the reliability of the SNV-CARS LS-SVM model. The optimal model was the NOR-CARS preprocessed LS-SVM model, which achieved $R_c^2$ and $R_p^2$ values of 0.9953 and 0.9880, respectively. Additionally, the RMSEC and RMSEP values were 0.0195 and 0.0303, respectively. Moreover, the RPDc and and RPDp values were 14.5494 and 8.9784, respectively (Fig. 6).

Moving on to the modeling and validation of delphinidin, it was found that the predictions from the PLSR model were unreliable. Applying preprocessing techniques such as SNV, AUT, NOR, SG, and MC did not improve the accuracy rate of the predictions (Fig. 7).

In contrast, for the LS-SVM model, the SNV LS-SVM also exhibited a perfect accuracy rate, but caution must be exercised regarding its reliability. The NOR LS-SVM model yielded the highest accuracy rate, with an $R_c^2$ value of 1.000 and $R_p^2$ value of 0.9967. Moreover, the corresponding values for RMSEC and RMSEP were notably low, measuring 0.0000 and

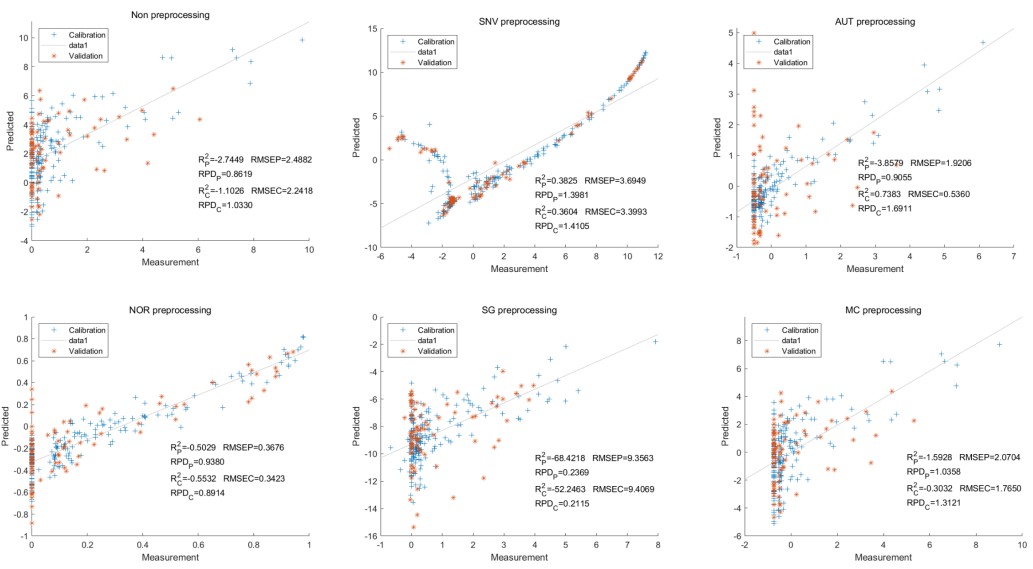

**Figure 4** Prediction results of cyanidin content in PLSR models based on all-band.

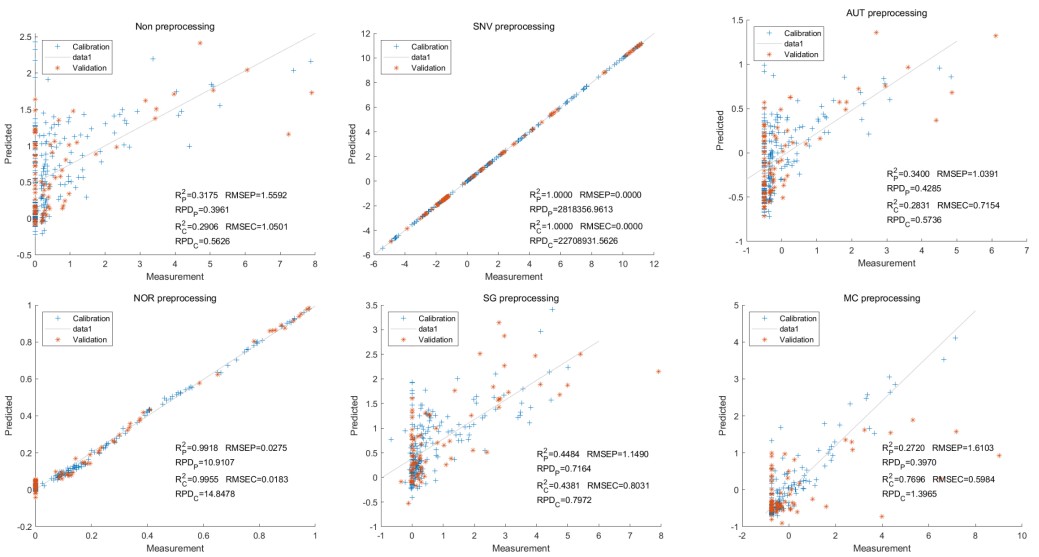

**Figure 5** Prediction results of cyanidin content in LS-SVM models based on all-band.

0.0203, respectively. Additionally, the RPDc and and RPDp values were 31,992.5331 and 17.7166, respectively (Fig. 8).

Furthermore, for the NOR-CARS LS-SVM model, the predicted result achieved an $R_c^2$ value of 0.9997 and $R_p^2$ value of 0.9959 The corresponding values for RMSEC, RMSEP

Peer J

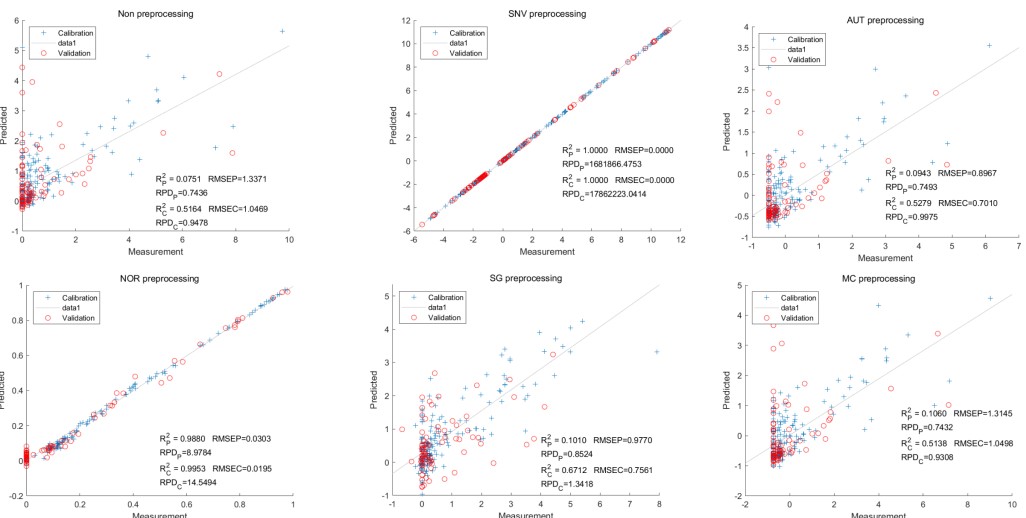

**Figure 6** Prediction results of cyanidin content in LS-SVM models based on CARS extracted feature variables.

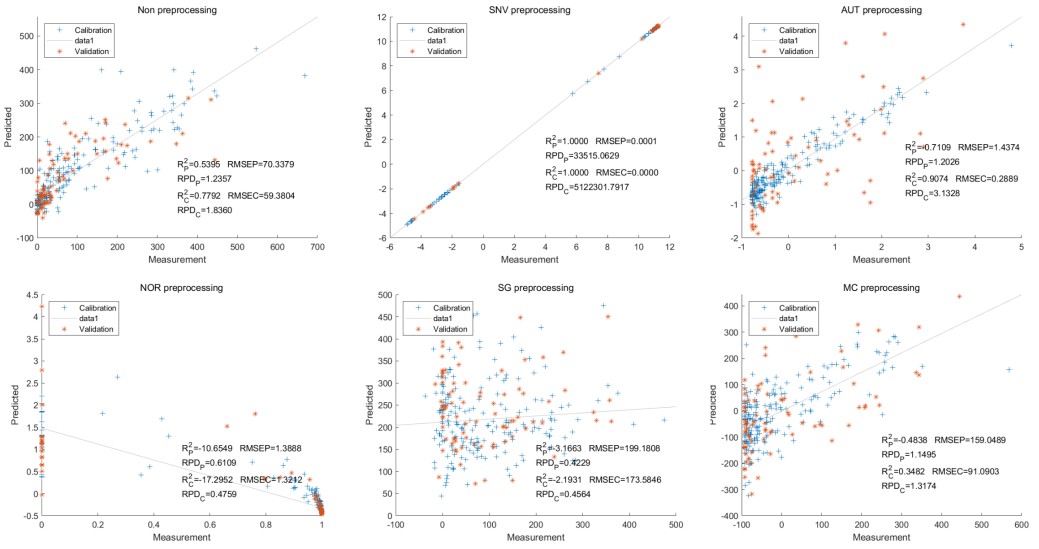

**Figure 7** Prediction results of delphinidi content in PLSR models based on all-band.

were notably low, measuring 0.0061, 0.0210, respectively and the the RPDc and and RPDp values were 57.9594 and 15.3560, respectively (Fig. 9).

Then, in the modeling and validation of petunidin, the PLSR analysis indicated that all of the predicted outcomes were not satisfactory. The SNV PLSR model displayed the highest level of accuracy, with an $R_c^2$ value of 0.6938 and $R_p^2$ value of 0.5935. However, the RMSEP
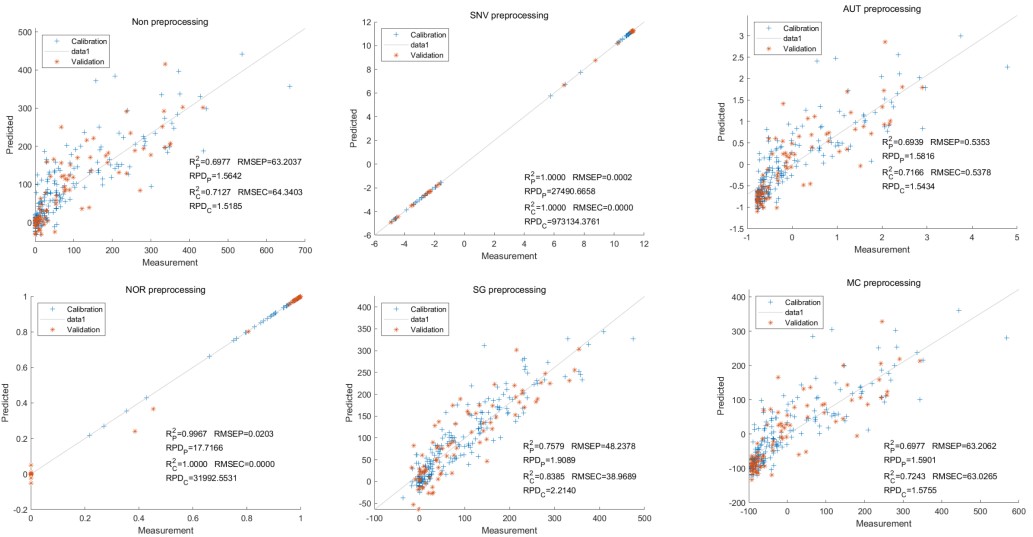

**Figure 8** **Prediction results of delphinidi content in LS-SVM models based on all-band.**

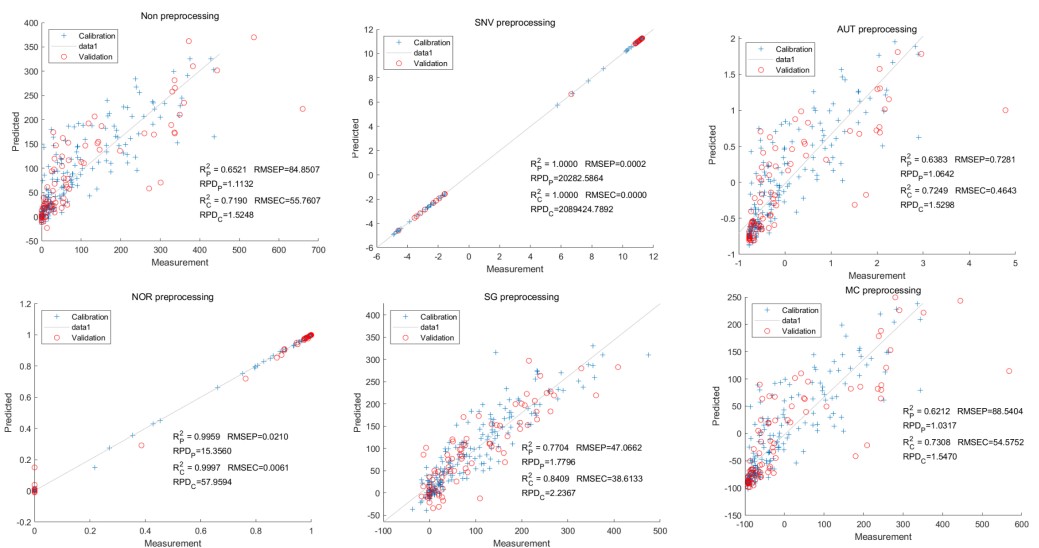

**Figure 9** **Prediction results of delphinidi content in LS-SVM models based on CARS extracted feature variables.**

and RMSEC values for this model were 2.7460 9 and 2.7413 respectively. Additionally, the RPDc and and RPDp values were 1.9109 and 1.6605, respectively (Fig. 10).

Similarly, the SNV LS-SVM model exhibited a perfect accuracy rate in its predictions, although caution should be exercised regarding their reliability. The NOR LS-SVM model had the highest accuracy rate, with an $R_c^2$ value of 0.9979 and $R_P^2$ value of 0.9977. Furthermore, the RMSEC and RMSEP values for this model were 0.0141 and 0.0158

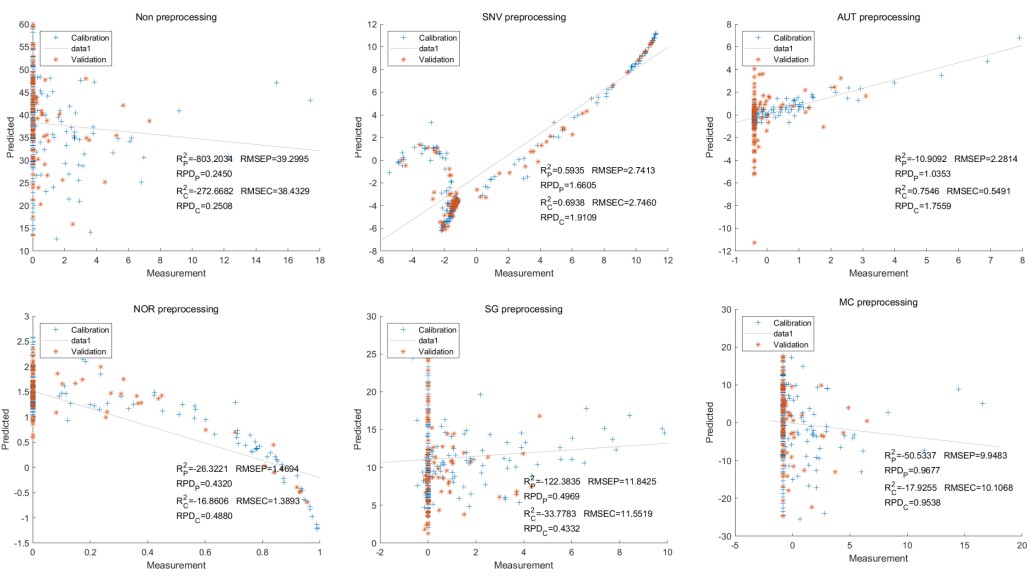

**Figure 10 Prediction results of petunidin content in PLSR models based on all-band.**

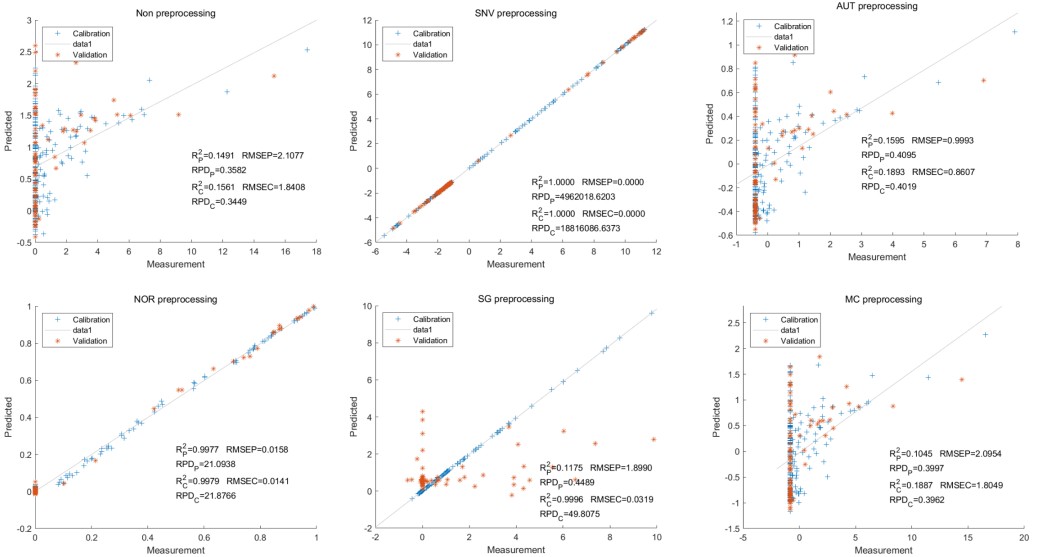

**Figure 11 Prediction results of petunidin content in LS-SVM models based on all-band.**

respectively. Additionally, the RPDc and and RPDp values were 21.8766 and 21.0938, respectively (Fig. 11).

Additionally, the NOR-CARS LS-SVM model produced an $R_c^2$ value of 0.9894. and $R_p^2$ value of 0.9889, with corresponding RMSEC and RMSEP values of 0.0339 and

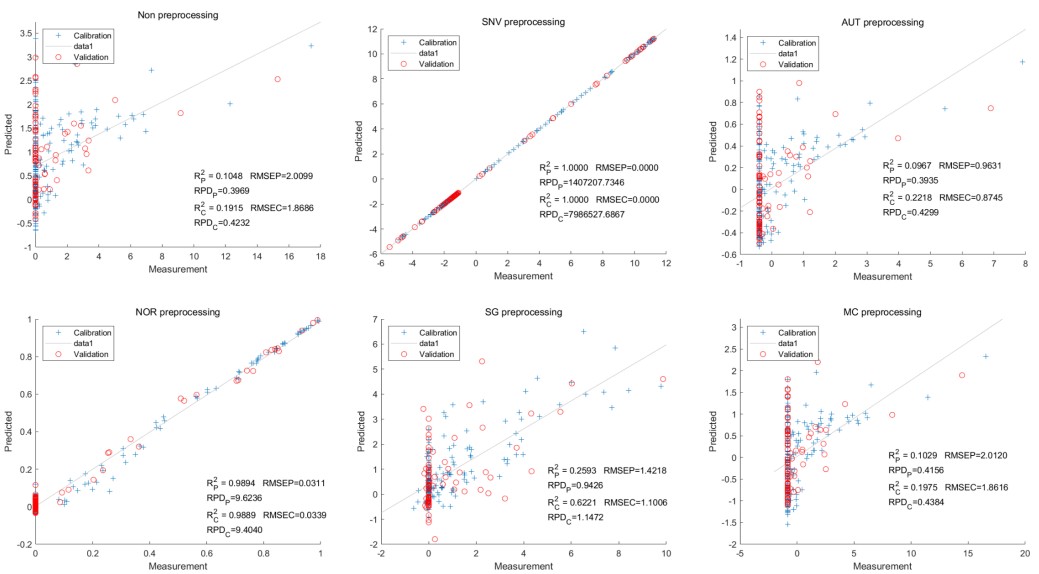

**Figure 12 Prediction results of petunidin content in LS-SVM models based on CARS extracted feature variables.**

0.0311 respectively. Additionally, the RPDc and and RPDp values were 9.4040 and 9.6236 respectively (Fig. 12).

Lastly,the results of the PLSR analysis indicated that all of the predicted outcomes for total anthocyanin were not satisfactory(Fig. 13). Similarly, the SNV LS-SVM model displayed a high level of accuracy in its predictions. Nevertheless, these results should be interpreted with caution, as they may not be entirely reliable. The NOR LS-SVM model produced the highest accuracy rate, with an $R_c^2$ value of 1.0000 and $R_p^2$ value of 0.99673. The corresponding RMSEC and RMSEP values were 0.0009 and 0.0161 respectively. Additionally, the RPDc and and RPDp values were 375.334 and 19.2105 respectively (Fig. 14).

Furthermore, the NOR-CARS LS-SVM model yielded promising predicted results. Its $R_c^2$ value of 0.999 and $R_p^2$ value of 0.997. The corresponding RMSEP and RMSEC values were 0.0036 and 0.0202 respectively. Additionally, the RPDc and and RPDp values were 91.0856 and 18.4801 respectively (Fig. 15).

## DISCUSSION

The color of eggplant fruit is determined by the type and content of anthocyanins and chlorophyll (*Liu et al., 2015*). The perception of fruit color by the human eye is influenced by various factors. In the present study, two eggplants were observed to have a white color, but they contained a minimal amount of delphinidin. Therefore, it is important to measure the content and type of anthocyanins using appropriate instruments and equipment.

HPLC has been widely used for detecting the content and type of anthocyanins in eggplant (*Todaro et al., 2009*; *Ferarsa et al., 2018*). The HPLC results of this study revealed that out of the 8 white and 28 green eggplants tested, none contained any type

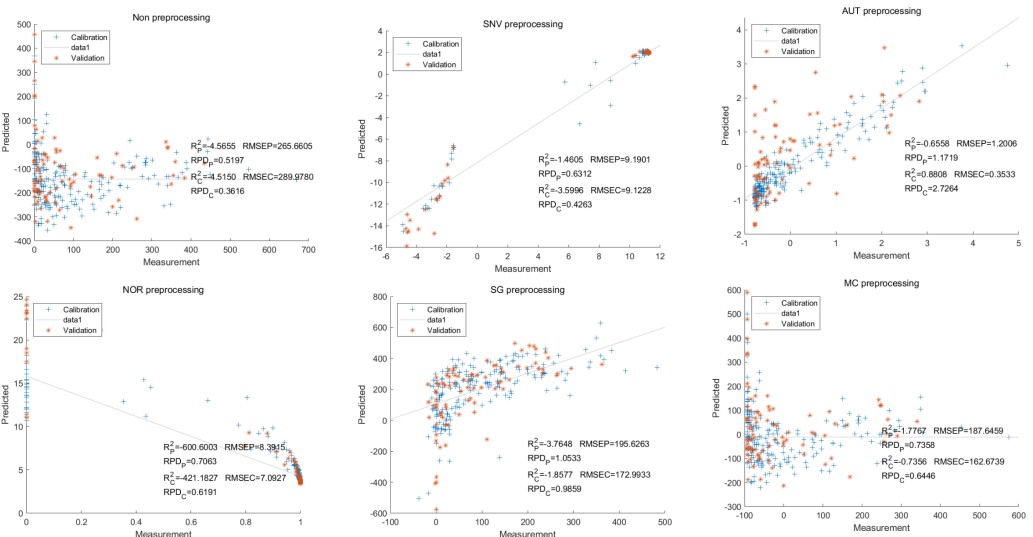

**Figure 13** Prediction results of total anthocyanin content in PLSR models based on all-band.

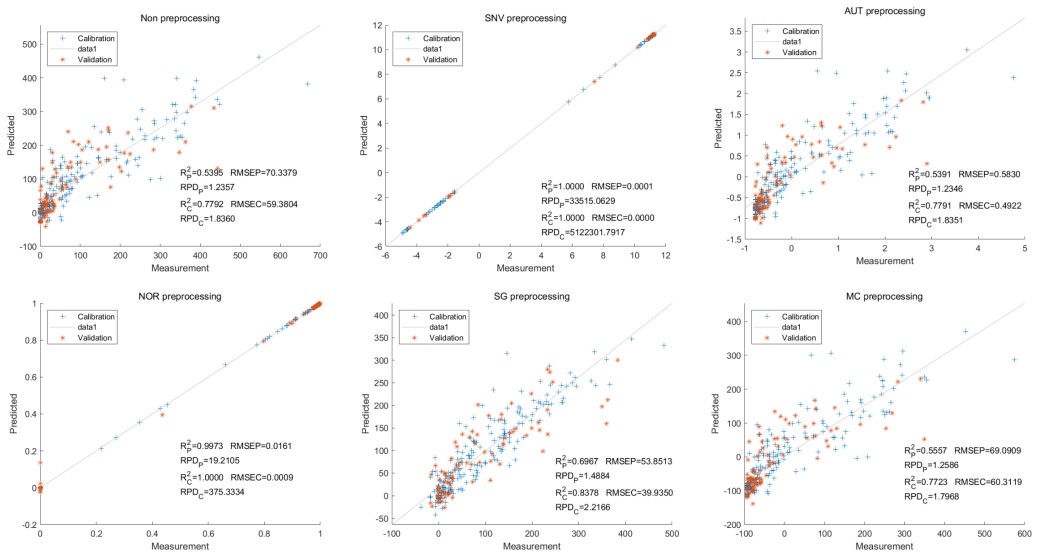

**Figure 14** Prediction results of total anthocyanin content in LS-SVM models based on all-band.

of anthocyanin. Although eggplants exhibit a variety of colors, to the best of our best knowledge, only petunidin, delphinidin, and cyanidin have been reported in eggplant peel (*Basuny, Arafat & El-Marzooq, 2012*; *Niño Medina et al., 2017*; *Todaro et al., 2009*). Pelargonidin, peonidin, and malvidin have not been reported, which is consistent with our results (*Basuny, Arafat & El-Marzooq, 2012*; *Niño Medina et al., 2017*; *Todaro et al., 2009*). However, it should be noted that this study focused on purple long eggplants. Further research is needed to determine if other types or genotypes of eggplant contain pelargonidin, peonidin, and malvidin.

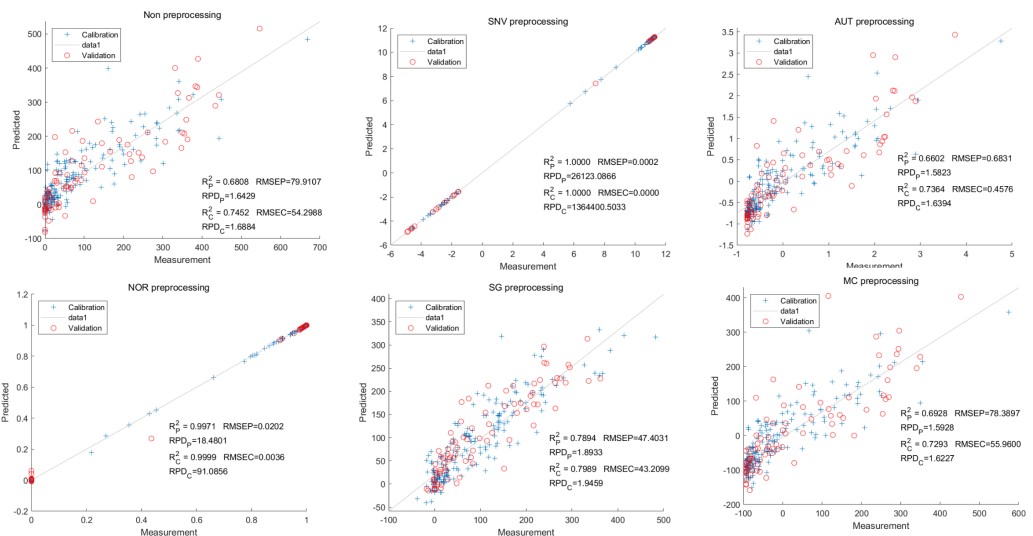

**Figure 15  Prediction results of total anthocyanin content in LS-SVM models based onCARS extracted feature variables.**

Hyperspectral imaging detector primarily detect the specular and scattered waves, while the secondary metabolites demonstrate distinctive absorption peaks (*Sarić et al., 2022*). In particular, the absorption peaks of chlorophyll were observed at approximately 500 nm and 700 nm. Consequently, the green and green-purple eggplants exhibited lower reflectance levels around these wavelengths. Additionally, the green-purple eggplant contains anthocyanin, resulting in even lower reflectance compared to the green eggplant. On the other hand, delphinidin displayed absorption peaks around 550 nm, leading to reduced reflectance at this specific wavelength for light-purple eggplants. It is widely recognized that higher color intensity corresponds to a lower reflectance and darker shades exhibit a lower reflectance. Therefore, the white eggplant demonstrated the highest reflectance values, while the dark-purple eggplant displayed the lowest reflectance ranging from 400 nm to 700 nm. The variation in reflectance among different colored eggplants serves as the basis for the non-destructive detection of eggplant peels through hyperspectral imaging.

The reflectance spectral data still contain background interference and noise caused by the current of the hyperspectral system. It is necessary to pretreat the spectral data to minimize background interference and improve model prediction accuracy (*Liu et al., 2015*). However, it is uncertain which method will yield the best result (*Zhang et al., 2017*). In the present study, five pretreatment methods were applied to the spectral data and the content of anthocyanins. The NOR pretreatment method yielded the best result for the LS-SVM model.

The PLS regression model has been widely utilized in hyperspectral analysis as it relates independent variables to an integer representing the sample class (*Burnett et al., 2021*; *Chen et al., 2015*; *Pandey et al., 2017*; *Zhang et al., 2022*). The LS-SVM has the advantages of speed and good generalization ability for regression (*Mehrkanoon & Suykens, 2012*). *Zhang et al.*

*(2017)* and *Chen et al. (2015)* showed that support vector regression (SVR) models behave globally better than PLSR for the estimation of anthocyanin in wine grapes. Overall, the PLSR and LS-SVM models showed varying levels of accuracy in our predictions, with the NOR LS-SVM model consistently outperforming the others.

Although NOR LS-SVM models yielded ideal predicted results, most of the models exhibit scattered points throughout the figure. Certain parameters show a high concentration of points in a small region, with only a few points distant from this cluster. The lower $R^2$ value and high RMSE value indicate that the majority of models were not ideal. This lack of accuracy may be attributed to significant variations in anthocyanin content, which impairs the predictive ability of the models. Additionally, the average reflectance of white and dark-colored eggplants differs significantly, further diminishing the models' predictive ability. Consequently, the NOR and SNV preprocessing methods yielded more ideal prediction outcomes. The presence of chlorophyll in green and green samples, which may not directly correlate with anthocyanin content, limits the applicability of prediction models to all samples.

Previous research has shown that the biosynthesis pathway of anthocyanins in eggplant peel is similar to that of other crops, involving multiple genes involved in the biosynthesis and regulation of eggplant anthocyanins (*Zhang et al., 2014*). Recently, researchers have conducted QTL mapping based on visual discrimination. Several transcription factors that regulate eggplant anthocyanin synthesis have been cloned by *Guan et al. (2022)*, *You et al. (2022)*, *Zhang et al. (2014)*, and *Zhou et al. (2020)*. However, these transcriptional regulations only determine whether eggplants can synthesize anthocyanins or not, without explaining how eggplants utilize the same substrate to produce different types of anthocyanins. The relative lag in related research is due to the inability to accurately determine the types and contents of anthocyanins during phenotype identification. Our study addresses this gap by establishing a non-destructive detection method for different types of anthocyanins in eggplant peel, providing a viable approach to QTL mapping of eggplant anthocyanin biosynthesis. We have already constructed the relevant mapping population and will employ the models developed in this study for QTL mapping, aiming to enrich and elucidate the biosynthetic mechanisms of anthocyanins in eggplant peel.

## CONCLUSIONS

In this study, 20 different varieties of eggplant were selected and we utilized the SVN, AUT, NOR, SG, and MC methods to preprocess the hyperspectral reflected data. Additionally, we used the CARS method was used to screen out feature variables. The PLSR and LS-SVM models were applied to predict the anthocyanin content in the eggplant peel. Notably, the NOR-CARS LS-SVM yielded the best results, with an $R_p^2$ and $R_c^2$v alue exceeding 0.9000 for cyanidin, petunidin, delphinidin, and total anthocyanin. These findings suggest that the combination of hyperspectral imaging and NOR-CARS LS-SVM enables fast,

non-destructive, and high-precision detection of anthocyanin content. This advancement will greatly benefit eggplant breeding.

### Funding

This research was funded by Natural Science Foundation of Hai Nan province (Grant No. 320RC700 and 322QN374), Key R&D Projects in Hainan Province (Grant No. ZDYF2023XDNY041), Central Public-interest Scientific Institution Basal Research Fund (Grant No. 1630062022003), Key R&D Projects in Guangdong Province (Grant No. 2022B0202080003). The funders had no role in study design, data collection and analysis, decision to publish, or preparation of the manuscript.

### Grant Disclosures

The following grant information was disclosed by the authors:
Natural Science Foundation of Hai Nan Province: 320RC700, 322QN374.
Key R&D Projects in Hainan Province: ZDYF2023XDNY041.
Central Public-interest Scientific Institution Basal Research Fund: 1630062022003.
Key R&D Projects in Guangdong Province: 2022B0202080003.

### Competing Interests

The authors declare there are no competing interests.

### Author Contributions

- Zhiling Ma performed the experiments, prepared figures and/or tables, authored or reviewed drafts of the article, and approved the final draft.
- Changbin Wei performed the experiments, authored or reviewed drafts of the article, and approved the final draft.
- Wenhui Wang performed the experiments, authored or reviewed drafts of the article, and approved the final draft.
- Wenqiu Lin conceived and designed the experiments, analyzed the data, authored or reviewed drafts of the article, and approved the final draft.
- Heng Nie performed the experiments, analyzed the data, prepared figures and/or tables, authored or reviewed drafts of the article, and approved the final draft.
- Zhe Duan performed the experiments, prepared figures and/or tables, authored or reviewed drafts of the article, and approved the final draft.
- Ke Liu performed the experiments, authored or reviewed drafts of the article, and approved the final draft.
- Xi Ou Xiao conceived and designed the experiments, prepared figures and/or tables, authored or reviewed drafts of the article, and approved the final draft.

### Data Availability

   The raw measurements are available in the Supplementary Table.

## Supplemental Information

Supplemental information for this article can be found online at http://dx.doi.org/10.7717/peerj.17379#supplemental-information.

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
