# Peer review of "Non-destructive prediction of anthocyanin concentration in whole eggplant peel using hyperspectral imaging"

_PeerJ, doi:10.7717/peerj.17379_

## Round 0.1 · original submission · Major Revisions

Dear authors

You should improve your manuscript by taking into account the reviewers' comments. In particular, consider the material and method section in a more descriptive manner.

Best regards

Reviewer 1 ·

Basic reporting

The manuscript exhibits clear and unambiguous language, maintaining a professional tone throughout. Adequate literature references are provided, and the manuscript establishes a solid background and context within the field. The structure of the article adheres to professional standards, including well-designed figures and tables.

Experimental design

The research question is clearly defined, demonstrating its relevance and significance. The manuscript explicitly outlines how the research addresses a recognized knowledge gap, emphasizing the contribution to the existing body of knowledge.

Validity of the findings

The presented study on hyperspectral imaging for anthocyanin content analysis in eggplant peel is commendable for its practical implications in eggplant breeding. However, the impact and novelty of the research have not been explicitly assessed. It would be valuable for the authors to address the broader significance of their findings within the field and highlight any innovative aspects that contribute to existing knowledge.

Additional comments

The study on hyperspectral imaging for anthocyanin content analysis in eggplant peel is a noteworthy contribution to the field. The inclusion of 20 eggplant varieties with diverse colors provides a comprehensive exploration of anthocyanin variations, enhancing the study's applicability to real-world breeding scenarios.

However, to strengthen the manuscript, it would be beneficial to elaborate on the practical implications of the findings for eggplant breeding. Exploring how the identified anthocyanin variations could influence breeding strategies or improve the nutritional quality of eggplants would add depth to the study. Additionally, considering the potential limitations of hyperspectral imaging, such as sensitivity to environmental conditions, would contribute to a more comprehensive discussion.

Furthermore, providing insights into the scalability and feasibility of implementing hyperspectral imaging in real-world breeding programs would enhance the practical significance of the research. Overall, while the study is well-executed and provides valuable insights, addressing these additional aspects would further enrich its impact and relevance in the field of eggplant breeding and agricultural research.

·

Basic reporting

The research is generally understandable, and sufficient language has been used to express the aim to be achieved clearly. In order to support the subject, the literature was selected from current and up-to-date articles. The introduction section, which covers the entire subject, strengthens the research structure and creates a solid basis for revealing the differences.

Due to the article's structure, it has content that can meet PeerJ standards. On the other hand, the presentation of data could be more orderly. References to the applied methods are not given, and no connection is established between the research findings and the tables and figures given to facilitate understanding of the subject.

There are errors in the in- and end-of-text reference notation, depending on the journal format.

Experimental design

Anthocyanin extraction is explained in the method section. It is stated that anthocyanin extraction was done according to the “High Performance Liquid Chromatography (HPLC)” method. However, there is no explanation or reference to the method.

No reference is given to HPLC analysis. The feature variables extraction process was stated to be done according to the Competitive adaptive reweighted sampling method, but no explanation or reference was given about the procedure. The method section needs to be completed in its current state.

It is stated that the model algorithms were made using least squares support vector machine (LS-SVM) and partial least squares regression (PLSR) with MATLAB 2022a software. It will be useful to present the process steps of the algorithm used in a flowchart to evaluate the fiction.

It was stated that 20 eggplant varieties in different colors were selected in the study, and a total of 277 eggplants with different colors such as white, green, light purple, green-purple, and dark purple were evaluated. Additionally, the HPLC chromatogram and spectral reflection of eggplant peel in different colors are given in Figure 1. Differences in reflections, especially in the visible region, have been associated with the presence of anthocyanin, although this is not fully known. The accuracy of this statement is questionable. First of all, eggplants of different colors should be grouped and a repeated comparison should be made. There is no mention of any trial plan or repetition here. In research conducted on plants, changes in plant physiology are generally observed in NIR (700-750 nm) and Red-Edge (750-900 nm) bands. It is necessary to better examine whether these differences in the Blue-Green band around 500 nm are due to eggplant skin color differences or anthocyanin differences.

Validity of the findings

Page 8, Line 162-163, "Light-purple and dark purple eggplants exhibit a minimum reflectance around 500 nm, which is associated with higher anthocyanin content and lower reflectance (Figure 1C)".
 Figure 1C shows the reflectance values of eggplants according to each color. On the other hand, no data is showing the relationship between color and anthocyanin content.

Page 8, Line 169-170, “Moreover, pretreating spectral data has the potential to augment the amount of feature 170 variables (Table S2 FigureS1-S4).”
 It is not clear what is meant by Table S2 Figure S1-S4. Such a definition is not included in the Tables and Figures.

Page 9, Line 183-184, “Matlabe software”

 It should be corrected using Matlab software.

Page 9, Line 218-219, The results of the PLSR analysis indicated that for all the predicted results models tested, the SNV PLSR model performed the best, with an R2 value of 0.9149.
 R2 value should be corrected to 0.9883.

Page 17, Table 2. Preprocessing of Spectral Data with different algorithms.

 It needs to be made clear which criteria were considered in revealing the differences in the average reflectance values at different wavelengths, which were not pre-processed and were evaluated based on the average reflectance spectra. While the wavelength values generally follow a collective path, what do the values that differ from this structure mean? More explanation is needed!!!

Additional comments

Although this research aimed to estimate the anthocyanin concentration in eggplant peel using hyperspectral imaging, the results show that anthocyanin and its six components can be statistically separated. However, this only partially meets the statement stated in the title and purpose of the article. It is stated that the separation is made according to the wavelength-average reflection graphs given in Table 2. There is no value in the table indicating which deviation corresponds to what amount of anthocyanin. Moreover, the article does not include a numerical equivalent of the variability in the amount of anthocyanins according to colors. Although it is not fully explained, there is only a connection between colors and the amount of anthocyanins, and it needs further explanation.

Reviewer 3 ·

Basic reporting

Non-destructive prediction of anthocyanin
concentration in whole eggplant peel using
hyperspectral imaging (#92541)
The work presents an interesting demand from the agricultural sector. The methods are properly applied, and the main novelty is the comparison of different statistical methods for data analysis. There are two major concerns that must be addressed by the authors, as presented below.
Since the problem proposed is related to samples with different colors, the authors must provide information to support the application of a hyperspectral imaging technique, which is much more expensive than a simple RGB imaging system or a colorimeter, which could probably classify the samples with high accuracy.

Experimental design

Appropriate number of samples, but it must be clarified the number of samples for each class, and the models could be calibrated independently to observe variations

Validity of the findings

It can be seen from figures 3-14 that most of the models have points scattered all over the figure, and that some parameters have a high concentration of points in a small region, with few points distant from this cluster. This greatly impairs the prediction model, and may provide illusory R2, demanding other features such as RPD and RER to discuss the models.
The authors must further discuss such phenomena, and actually suggest other problems that may have caused such bad prediction. Perhaps having samples that are white and dark, as observed in the spectral information, may not make possible to provide prediction models applicable to all samples, as chemical composition may vary not directly in accordance with anthocyanin content.

Additional comments

None

---

## Round 0.2 · Minor Revisions

You have attempted to revise your manuscript based on the reviewers's comments. However, there are still points that are not clarified. You have to reconsider it. You should specifically address reviewer 2's concerns. Maybe you should also consider revize the title of your manuscript. Additionally, your manuscript contains many typographical errors. Make it suitable for PeerJ format. Writing style, bibliography display in the text, etc. For example, in line 261: [12, 15] is an incorrect representation. Make it appropriate. Correct different font styles to Times New Roman.

"Although the article title states, "Hyperspectral imaging estimation of anthocyanin concentration in eggplant peel," it states that there is no relationship between color and the number of anthocyanins. In the article, it is necessary to specify in more detail the factor based on which a prediction is made from hyperspectral images. In other words, the hypothesis of the article could be clearer."

Reviewer 1 ·

Basic reporting

no comment

Experimental design

no comment

Validity of the findings

no comment

Additional comments

I am satisfied with the response of the authors.

·

Basic reporting

The issues mentioned in the first section of the article in the first report should be stated in this report to avoid repetition.

Inaccuracies in the text and references section that did not comply with the journal format have been corrected.

Experimental design

A reference to "High-Performance Liquid Chromatography (HPLC)" has been added to the article.

The citation form of the very long reference can be shortened to "HPLC, 2014" instead of "Ministry of Agriculture 133 of the People's Republic of China, 2014". The same correction should be made in the references list.

The flow chart of the PLSR method evaluated with MATLAB 2022a software is added to the article.

Although Table S1 and Table S2 are mentioned in the article, they do not appear in the Table section. It needs to be made clear which tables the statements S1 and S2 refer to.

Validity of the findings

Page 8, Line 162-163, "Light-purple and dark purple eggplants exhibit a minimum reflectance around 500 nm, which is associated with higher anthocyanin content and lower reflectance (Figure 1C)".
- Suppose there is no relationship between color and anthocyanin amount. In that case, further explanation is needed as to which factor is associated with hyperspectral images and how the graphics in the article were created. In other words, which feature is related to the amount of anthocyanin on hyperspectral images should be stated.

Page 8, Line 169-170, “Moreover, pretreating spectral data has the potential to augment the amount of feature 170 variables (Table S2 FigureS1-S4).”
- Table S2 still needs to be added to the text and tables index. Table 2 should either be added or clarified as to which table or figure is referred to avoid conceptual confusion.

Page 9, Line 183-184, “Matlabe software”

- The wording has been corrected. However, there are two separate designations, MATLAB 2022a and Matlab 2020a. These should be converted into a single format, "Matlab 2020a".

Page 9, Line 218-219, The results of the PLSR analysis indicated that for all the predicted results models tested, the SNV PLSR model performed the best, with an R2 value of 0.9149.
- The necessary correction has been made.

Page 17, Table 2. Preprocessing of Spectral Data with different algorithms.

- Table 2 still needs to be added to the text and/or tables index. Table 2 should either be added or clarified as to which table or figure is referred to avoid conceptual confusion!!!

Additional comments

"Table 2." It is also stated in the answer given by the author that it is not included in the article. However, no corrections or additions were made. This issue needs to be clarified.

Reviewer 4 ·

Basic reporting

The language of the article is professional and written in proper English. The discussed literature and references are adequate and the manuscript provides beneficial information for the field of expertise. The hypothesis was created clearly and the results are relevant to the hypothesis.

Experimental design

The experiment is well-designed and proper methods were applied. I suggest authors measure and add the light intensity while hyperspectral imaging to ensure the repeatability of the experiment. It is known that the reflectance values of the objects may greatly vary under different light intensities. It is enough to measure the same environment again and add the result in the method section, where the light source is explained in the text. The statistical approach is appropriate for the data.

Validity of the findings

The study has successful and useful results so I believe this article will provide beneficial information to the area of expertise. The underlying data was provided. The conclusion is linked to the hypothesis and the study has original results.

---

## Round 0.3 · Minor Revisions

Your manuscript still needs minor corrections identified by the reviewers.

Reviewer 3 ·

Basic reporting

The authors have addressed all suggestions.
As requested, RER is ratio between the range of the reference data to RSEP, and indicates how the range of the reference values impact the error.
Also, although the authors state that the objective is not to measure color, it is stated that there is a correlation between color and anthocyanin content and type. So perhaps the authors may present this correlation on the work, to provide support for future works.

Experimental design

Why not having a balanced dataset among all types of samples? The authors should discuss this issue based on prediction of each individual class

Validity of the findings

Please see previous comments. I suggest discussing the boundaries of this work based on those comments

Reviewer 4 ·

Basic reporting

The authors used proper language in the manuscript. The literature and references are sufficient for the subject. All tables, figures, and raw data indicate the results clearly and relevant to the hypothesis.

Experimental design

The experimental design, statistical approach, and presentation of the results are proper for publication in PeerJ. The authors clearly defined the purpose of the research and presented relevant results to the aims. They used a good technique and generated useful information for breeding studies of eggplant.

Validity of the findings

I believe the study findings will be useful for eggplant breeding research. The technique could help as a rapid and objective tool at the selection level in eggplant breeding. Therefore I believe this study will have a positive impact on further studies of eggplant breeding. Conclusions are well-linked and limited to the research questions.

Additional comments

I checked the editings of the authors and could state that the authors made corrections and additions to the text by considering the reviewer's suggestions.

---

## Round 0.4 · accepted · Accept

Congratulations. By meeting the reviewers' and editorial corrections, your manuscript was accepted.